# Correlation of Adiponectin and Leptin with Anthropometrics and Behavioral and Physical Performance in Overweight and Obese Chinese College Students

**DOI:** 10.3390/biology13080567

**Published:** 2024-07-27

**Authors:** Jingyu Sun, Jiajia Chen, Antonio Cicchella

**Affiliations:** 1Sports and Health Research Center, Department of Physical Education, Tongji University, Shanghai 200092, China; jysun@tongji.edu.cn (J.S.); tjchenjiajia@163.com (J.C.); 2International College of Football, Tongji University, Shanghai 200092, China; 3Department for Quality of Life Studies, Bologna University, 40127 Bologna, Italy

**Keywords:** leptin, adiponectin, obese Chinese college students

## Abstract

**Simple Summary:**

We studied a population of overweight and obese Chinese university students. Obesity, which was previously limited to the Western world, has emerged in Asia and China in recent years. Owing to the massive population of the Asian region, even effects on a small percentage of this population correspond to substantial problems. University students are an at-risk group for obesity due to the sedentary lifestyle imposed by their heavy study schedule. We measured the physical capacities, as well as several biochemical and behavioral variables, of Chinese students. We measured the blood levels of two multifunctional hormones, namely leptin (LEP) and adiponectin (ADPN), and levels of physical activity. We found that in obese students, physical capacities are linked with the hormone ADPN, which has been hypothesized to be linked with muscle development. We also found that sleep was associated with the blood levels of LEP and ADPN. To the best of our knowledge, our study is the first work to provide data on the association of LEP and ADPN with behavioral, physical, and biochemical variability in overweight and obese Asian populations. Such an association has potential for use as a comparison tool.

**Abstract:**

The aim of this study is to assess the relationship of leptin (LEP) and adiponectin (ADPN) with other circulating fat markers, physical capacity, behaviors, and anthropometric indices in a population of overweight and obese Chinese university students. LEP and ADPN levels, as well as behavioral, anthropometric, biochemical, and performance characteristics, were measured. Method: A total of 17 anthropometric parameters, 8 questionnaires (investigating quality of life, sleep, eating, perceived functioning, stress, and depression), 9 biochemical parameters, and 12 functional parameters were investigated. Results: In contrast to previous studies, our work found an unusually strong relationship between LEP and ADPN (r = 0.961, *p* = 0.000) that can be related to ethnicity. We also found that LEP and ADPN were associated with stress and bodily pain. A total of 12 anthropometric measures were also associated with LEP/ADNP levels. Moreover, LEP and ADPN were found to be related to lower limb, hand, and abdominal strength; blood pressure; and basic metabolism. However, we did not find associations with sleep; eating habits; or cardiovascular fitness, which was measured in the form of resting heart rate and VO_2max_. Conclusion: This study reveals new relationships of LEP and ADPN with selected anthropometric and behavioral parameters in obese Chinese college students.

## 1. Introduction

Leptin [1,2] (LEP) and adiponectin [3] (ADPN) are two adipocytokines that have been widely studied in connection with obesity [4]. LEP increases and ADPN decreases with increasing body fat. The ADPN:LEP ratio is a prognostic factor for the development of obesity [5], cardiovascular disease [6], insulin resistance [7], and diabetes [8], as well as several other conditions, due to the ubiquity of LEP and ADPN in the human body [1,2]. Given that they are produced by adipose tissues, plasma LEP and ADPN have been connected with different anthropometric measures in the Caucasian population, particularly with waist and hip circumference, waist:hip ratio, BF%, and body fat mass [9]. By contrast, in the Asian population, plasma LEP and ADPN are connected with abdominal fat [10,11,12,13,14]. The relationship of LEP and ADPN with anthropometric measures is of clinical interest because it may aid in the diagnosis of hyperleptinemia, which has been widely explored [15,16]. Considering the proximity of LEP receptors to neuropeptide Y receptors in the hypothalamus [17], LEP has been studied in connection with anxiety, stress [18,19,20], depression [21,22,23], sleep [24], and eating behaviors [23,25]. LEP and ADPN levels have also been associated with behavioral factors, such as physical activity levels [25,26]. Although LEP and ADPN, as ergogenic molecules, have been suggested as markers of training status [27] in athletes, only a few studies on the possible link of LEP with the capacity for physical performance exist. In children, LEP is inversely correlated with the amount of physical activity [28] and positively correlated with VO_2max_ [29]. However, the relationships of LEP and ADPN with physical performance in obese adults have been poorly investigated [30]. To the best of our knowledge, no studies exist on the relationship of LEP and ADPN with basic physical performance (lower limb, hand, and abdominal muscle strength; balance; and flexibility) in overweight/obese Asian college students. Furthermore, studies investigating behavioral, physical, biochemical, and anthropometric indices in the same cohort of overweight and obese subjects do not exist.

As suggested by the literature [10,11,12,13,14], the associations of LEP and ADPN with anthropometrics in Asians are different from those in Caucasians. Comparing different populations within the same BMI range revealed that ADPN concentrations were lower in the Indonesian population than in the Dutch population [31]. In obese Caucasians, LEP and ADPN showed no or low correlations with each other [32]. LEP has been hypothesized to be connected with different behaviors [33]. It regulates food intake, reward, and motivation. Reduced LEP impairs cognition [33].

To the best of our knowledge, only a few studies have linked LEP and ADPN with behavioral characteristics in Chinese populations. Behavior and LEP levels have been studied in Chinese schoolchildren. LEP was found to be associated with low math scores and high verbal scores. Low ADPN was found to be correlated with family-related behavioral problems [34]. LEP has been proposed as a biomarker of sleep disorders given that it acts as a mediating factor between obesity and poor sleep [35]. High ADPN and low LEP levels were associated with anxiety [36] and stress [37]. Therefore, our aim is to assess the relationship of LEP and ADPN with other circulating fat markers, physical capacity, behaviors, and anthropometric indices in a population of overweight and obese Chinese university students.

## 2. Materials and Methods

### 2.1. Participants

This study was a randomized, controlled trial. Obese and overweight adults were recruited from Tongji University through workshops, leaflets, social media, official web pages, and paper bulletins disseminated on campus.

The ethics committee of Tongji University approved all interventions (tjdxsr046). A total of 40 respondents were invited to visit the research department. Prior to the collection of eligibility data, researchers carefully explained the purpose and protocol of the study in detail. Prior to registration, participants signed a written informed consent letter. All participants completed an initial screening by filling out questionnaires on background information (including brief sedentary questions, medical history, current health status, a food intake questionnaire [FFQ25 [38]], the International Physical Activity Questionnaire [IPAQ [39]], and a physical examination). A total of 8 participants did not meet the inclusion criterion of sedentary obesity, and out of the 32 participants who were invited to enroll in this study, 6 refused. This study included 26 subjects (3 females) from different colleges of Tongji University in Shanghai (age 19.23 ± 1.39 years) (Table 1). Obesity was defined in accordance with Chinese reference values [40] as BMI > 28.

### 2.2. Inclusion and Exclusion Criteria

The inclusion criteria were (1) being apparently sedentary in accordance with living a sedentary lifestyle (spending more than 8 h awake in a sitting position daily and continuously sitting still for over 90 min); (2) age between 18 and 25 years; (3) BMI ≥ 25; (4) has not engaged in regular exercise for more than 30 min for 3–5 months; and (5) IPAQ score < 600 metabolic equivalents·min/week. Out of 26 subjects, 15 were obese (BMI > 28).

The exclusion criteria included (1) severe cardiovascular or skeletal muscle disease; (2) diabetes complications (such as postexercise hypoglycemia or autonomic neuropathy or peripheral neuropathy); (3) diagnosed with mental illnesses; and (4) other sports/exercise contraindications (diagnosed with heart problems). All the students regularly consumed their meals at the campus canteen.

### 2.3. Measurements

#### 2.3.1. Anthropometric Measures

A total of 16 anthropometric measures were taken in accordance with standard procedures [41]. In total, 11 circumferences (chest; minimum and maximal waist line; contracted and relaxed hip, thigh, calf, and biceps brachii), and 6 skinfolds (shoulder and tight crease, biceps, abdominal, iliac, and calf) were measured. Weight, fat, fat-free and muscle masses, and body fat% were measured with a X-SCAN PLUS (Selvas, Gyeong, Republic of Korea) bioelectric impedance scale. Height was measured by using a Hengkang Jiaye (Shenzhen, China) anthropometer. Physical tests, including 1 min sit-ups and squats, standing on one leg with eyes closed, and the sit-and-reach test, were performed in accordance with ACSM guidelines [42].

#### 2.3.2. Biochemical Parameters

Biochemical parameters were measured in duplicate after overnight fasting for at least 10 h. The following parameters were measured with the fully automated biochemical analyzer Dimension EXL 200, (Siemens Healthineers, Munchen, Germany): total cholesterol, triglycerides, high-density lipoprotein cholesterol, low-density lipoprotein, glucose, hemoglobin, and HbA1c. ADPN and LEP were quantified by using an E-EL-H0004-96 T human ADP/Acrp30 (ADPN) ELISA kit (BLKAZ5BYNR, Elabscience, Wuhan, China) and E-EL-H0113-96 T human LEP ELISA kit (EKF8PFBUAG, Elabscience, Wuhan, China) in accordance with the manufacturer’s instructions.

#### 2.3.3. Cardiorespiratory Indices

Basal VO_2_, VO_2max_, and basal metabolism (METS) were measured on the basis of the mean of the FatMax test results. FatMax is the optimal heart rate for fat utilization [43]. Subjects were asked to refrain from vigorous exercise for 24 h and fast for 10 h before the test. All FatMax tests were conducted in the morning between 8 a.m. and 12 a.m. to avoid circadian variance. A graded treadmill exercise test was used to perform the FatMax test. In brief, a walking warm-up was performed at 3.0 km/h with an incline of 1% for 3 min. Subsequently, the first stage of exercise was performed at a speed of 6.0 km/h for 2 min. Each increase in speed had a duration of 2 min. The protocol was modified to adapt to the physical capacities of the sample from a previous study employing FatMax on obese adults [44]. The speeds of the following steps were 7.0, 8.0, 9.0, and 10 km/h until the respiratory exchange ratio of 1.15 was reached or the subject became exhausted. VO_2_ and VCO_2_ were measured by using a portable metabolic system (K5, COSMED, Rome, Italy) and Polar H10 (Polar Electro, Tampere, Finland) heartrate band. The test was stopped immediately if the subject showed symptoms of dizziness, nausea, or dyspnea during the test. Vital capacity was measured with an electronic activity tester (Hengkang Jiaye, Shenzhen, China).

#### 2.3.4. Physical Performance Parameters

Diastolic and systolic blood pressure levels were measured with a standard sphygmomanometer, and resting heart rate was measured with a Polar H10 HR monitor. The grip strength of both hands was measured with a Jamar dynamometer. Progressive maximal contractions were performed for both hands, and the best value of 3 attempts was retained. All subjects were right-handed (Table 1).

#### 2.3.5. Scale Evaluation

SF36 [45] quality of life; the IPAQ scale [39], which was also validated for blood pressure [46]; HAMA for anxiety and depression [47,48] (mental and body anxiety and total scores); PSS for perceived stress levels [49]; SDS for anxiety [50]; TFEQR21 for eating behaviors [51,52]; FS14 [53] (physical and mental fatigue and total scores); and PSQI [54] for the assessment of sleep were used. The SF-36 measures 8 scales: physical functioning (PF), physical role, bodily pain (BP), general health, vitality, social functioning (SF), emotional role, and mental health. Questionnaires were submitted in the form of the Chinese version validated in previous studies [45,46,47,48,49,50,51,52,53,54]. All measurements were performed in the morning, and the testing order was randomized between subjects for each test by using a random number sorter.

#### 2.3.6. Statistical Analysis

Descriptive statistic and correlation analyses (Spearman) were performed by using SPSS IBM, SPSS Statistics 25. Levine and Shapiro–Wilk tests were used to check the homogeneity of variance and normality of data, respectively. All data were normally distributed. The extracted data included the mean and SD of participant characteristics, cardiorespiratory indices, and questionnaire scores.

## 3. Results

### 3.1. Sample Descriptive Statistics

Table 1 reports the means and standard deviations of the observed variables.

### 3.2. Questionnaires Scores

The questionnaire results of overweight and obese Chinese college students are reported in Table 2.

The mean PSQI score of our sample was within the range of normal sleep (0–5 points). However, we observed 10 subjects with a score between 6 and 10 (mild insomnia) and 1 subject with moderate insomnia (>10). The HAMA total cut-off point for anxiety disorders was 14. The mean of our subjects was below the anxiety threshold (6 subjects were beyond the threshold). The maximum FS14 fatigue score was 14. Our group average showed that our subjects, with the exception of 8 subjects who scored >10, did not perceive considerable fatigue. Our sample had a score of <50 on the SDS scale (no depression symptoms).

### 3.3. Correlations among Measured Variables

Table 3 reports the results of the correlation analysis. In contrast to several other studies [32,55,56,57], our work found that LEP and ADPN were strongly correlated with each other (r = 0.961, *p* = 0.000). As expected, LEP and ADPN were correlated with all other biochemical indices of fat metabolism that we measured (Table 3), with LEP showing an inverse correlation and ADPN a strong positive correlation with total cholesterol. LEP is correlated with LDL and HbA1c, and LEP and ADPN were correlated with HGB. In agreement with another study on obesity that found a CC of 0.670 (*p* = 0.0001) [58], this work discovered that ADPN was correlated with HDL (Table 3). ADPN showed a correlation with BMI; height; hip, bicep (contracted and relaxed), waist, and calf circumferences; shoulder crease skinfolds; and waist:hip ratio and muscle mass. LEP presented correlations with the previous variables, except for muscle mass, fat-free mass, waist circumferences, and shoulder crease skinfold. However, in contrast to ADPN, LEP showed a correlation with thigh circumference (Table 2). ADPN was correlated with muscle and fat-free masses, 1 min squat (reps), long jump, 1 min sit-up (reps), and hand grip but not with sit-and-reach test results and body balance. LEP exhibited the same correlations with physical performance (Table 2).

Although previous studies also observed a correlation between LEP and blood pressure in obese individuals with high blood pressure [59], our sample did not appear to have hypertension. METS was also correlated with LEP and ADPN. LEP was correlated with the total score on the PSQI questionnaire and with the PF and BP subscales of SF36. LEP and ADPN were correlated with the depression and stress scales, as well as with the BP and physical function scales (Table 3).

## 4. Discussion

Low (or decreased) levels of LEP and high (or increased) levels of ADPN are beneficial in terms of reducing weight and inflammation and desirable in overweight and obese individuals [3]. The present study aims to explore the relationships between LEP and ADPN in obese Chinese university students. Considering the absence of comprehensive studies exploring the relationships of LEP and ADPN with the anthropometric indices and behavioral and physical performances of Chinese college students, our present study is the first work to provide such information. LEP and ADPN are inversely correlated in obesity. The major finding of our study is a quasiperfect (r = 0.961, *p* = 0.000) correlation between LEP and ADPN. This result is surprising because LEP and ADPN have an inverse relationship in humans [32,55,56,57,60]. Given that we did not find any study on a similar cohort (age and ethnicity), we can hypothesize that this result can be attributed to diet. In fact, all of our subjects attend the same university and may share common dietary factors that influence ADPN levels. Although our sample size is relatively small, the subjects are highly homogeneous, all being college students of the same age and gender (with the exception of three females). They also consume their meals at the same place and sleep in dormitories under similar conditions. We can hypothesize that some factors can influence the strong correlation that we found between LEP and ADPN. For example, the consumption of green tea is common among Chinese students and could reduce LEP and increase ADPN considerably [61]. Moreover, the students live in Shanghai, wherein the local diet is rich in fish omega-3, which can also reduce LEP [62]. Another possible explanation for the strong correlation between LEP and ADPN may be the distribution of body fat in the Asian biotype [4]. Such a distribution is associated with the differing secretion of LEP and ADPN in the blood stream. Asians accumulate more visceral fat tissue and less subcutaneous fat than Caucasians. Although the association of LEP and ADPN with muscle mass (fat-free mass) and lower limb strength was previously observed in young tennis players [63] and Chinese high school students [64], it remains controversial. A previous work found that LEP was positively associated with muscle density (it would decrease skeletal muscle lipid content and promote lipid oxidation) [65], whereas another study discovered that LEP was negatively correlated with muscle density and relative hand grip strength [66]. In contrast to a study on young Caucasians [66] and Afro-Americans [67], our work found that hand strength was positively correlated with LEP. Our study also discovered a relationship between LEP/ADPN and abdominal strength. The correlation with muscle mass and thus with strength can be explained by the involvement of ADPN in the phosphorylation of adenosine monophosphate, which acts as a signaling factor in skeletal muscles [68].

We investigated the behavioral characteristics associated with LEP/ADNP. Although an association exists between feeding and LEP levels [69], we did not find such an association in our subjects. The relationship between high levels of LEP and sleep found in previous studies [70] could not be observed in our work. The LEP level in our sample was five times lower than that normally present in the obese Caucasian population [71]. Although a certain degree of this difference can be explained by the sensitivity of measurement methods, our Chinese samples had very low LEP levels. Our work found that LEP and ADPN were associated with depression and stress, confirming the findings in the Caucasian population [72]. Given that LEP and ADPN are also ergogenic [73], their association with the PF index measured by using the SF questionnaire is unsurprising. We hypothesize that the association with BP can be attributed to LEP-associated inflammation in obese individuals, as shown in a previous study [74]. However, the association of ADPN with pain remains under debate [75]. These factors could have contributed to the low levels of LEP and strong association between LEP and ADPN in our study. We discovered a correlation between ADPN and selected body dimensions (BMI; height; hip, biceps (contracted and relaxed), waist, and calf circumferences; shoulder crease skinfolds; waist:hip ratio and muscle mass). This result is in agreement with the hypertrophic role of ADPN [76]. In fact, ADPN was also correlated with muscle and fat-free masses, 1 min squat (reps), long jump, 1 min sit-up (reps), and hand grip but not with sit-and-reach test results and body balance. LEP presented correlations with previous variables, except for muscle mass, fat-free mass, waist circumferences, and shoulder crease skinfold. However, in contrast to ADPN, LEP showed a correlation with thigh circumference in agreement with the secretion of LEP by subcutaneous fat tissues [1]. A limitation of our study is that we did not measure visceral fat, which is highly correlated with LEP and inversely correlated with ADPN. In addition, diet was not assessed in detail, and we did not include a control group (which could have been helpful in strengthening the results). Our sample is also unbalanced in gender distribution. We are aware that our number of participants is limited and imbalanced. However, the incidence of obesity in China is considerably lower than in Western countries, and our sample included almost all of the obese students of the campus.

## 5. Conclusions

Our study found new relationships of LEP and ADPN with selected anthropometric and behavioral variables in obese Chinese college students. These correlations are different from those observed previously in the Caucasian population and deserve further investigation. We hypothesize that our results can be attributed to differences in diets. The relationships of LEP and ADPN with anthropometric and biochemical indices, performance, and mood indicate that LEP/ADPN has a relationship with selected anthropometric indices, mood states, and physical performance in overweight and obese Chinese college students.

## Figures and Tables

**Table 1 biology-13-00567-t001:** Anthropometrics, biochemical, cardiorespiratory indices, and functional measures (mean and standard deviation).

	Mean	SD	Range
Anthropometrics			
Height (cm)	176.92	±6.62	167.00–180.7
Weight (kg)	86.75	±20.46	65.60–142.10
BMI	29.28	±5.48	24.100–40.40
Fat-free mass (kg)	61.43	±13.60	41.30–87.90
Fat mass (kg)	26.75	±11.81	12.10–61.10
Muscle mass (kg)	58.4	±425	54.80–53.10
Body fat %	24.2	±6.53	17.30–30.30
Waist:hip ratio	0.94	±0.04	0.91–0.98
Chest circum (cm)	95.143	±19.73	101–123.00
Waistline (cm)	93	±13.12	78.50–129.00
Minimum waist circum (cm)	88.98	±10.16	73.20–116.00
Hipline (cm)	99.98	±20.87	91.00–106.00
Thigh circum (cm)	62.9	±7.16	53.00–78.00
Calf circum (cm)	41.75	±4.14	36.00–52.00
Biceps contract circum (cm)	32.07	±4.32	24.4–44.00
Biceps relaxed circum (cm)	30.91	±3.42	24.80–37.00
Abdominal skinfold (mm)	48.90	±19.64	19.50–75.00
Shoulder crease skinfold (mm)	33.89	±10.97	13.00–53.00
Biceps skinfold (mm)	39.51	±13.75	12.00–73.00
Thigh crease skinfold (mm)	50.14	±19.26	8.00–75.00
Iliac skinfold (mm)	24.53	±12.52	5.00–51.00
Calf skinfold (mm)	34.093	±13.81	12.00–56.00
Biochemical parameters			
TC (mM/L)	4.03	±0.92	1.87–5.67
TG (mM/mL)	1.14	±0.57	0.27–2.84
HDL (mM/L)	1.35	±0.25	0.79–1.85
LDL (mM/L)	2.59	±0.78	0.70–4.28
Glucose (mM/L)	4.76	±0.34	3.88–5.59
LEP (ng/mL)	3.83	±0.82	2.50–5.42
ADPN (mg/L)	8.69	±1.82	5.58–12.31
ADPN/LEP ratio	2.30	±0.11	2.07–2.48
HGB (mM/L)	82.66	±8.44	66.55–100.38
HbA1c (mM/L)	32.28	±16.14	19.72–98.49
Cardiorespiratory indices			
Basal VO_2_ (L/min)	1822	±157.04	1688–1995
VO_2_ max (L/min)	3374.27	±1016.87	1222–5889
VO_2_ max (mL/Kg/min)	40.46	±5.88	30.90–53.40
METS	11.71	±2.47	4.40–17.50
Physical performance parameters			
Systolic pressure (mmHg)	128.96	±22.35	95.00–157.00
Diastolic pressure (mmHg)	76.38	±10.01	63.00–97.00
Resting heart rate (beats/min)	80.44	±13.73	54.00–101.00
Hand grip strength (right hand) (kg)	37.64	±9.23	19.30–61.90
Vital capacity (mL)	3878.69	±913.54	2674–7140
Sit-and-reach (cm)	+14.05	±8.58	−2.90–(+)29.00
1 min sit-ups (number)	30.81	±6.22	18.00–45.00
Standing long jump (cm)	195.71	2 ± 5.71	144.00–243.00
1 min squats (number)	48.38	9 ± 0.71	31.00–65.00
Standing on one leg with eyes closed (s)	33.95	±32.37	1.20–131.53

BMI: body mass index; TC: total cholesterol; TG: triglyceride; HDL: high-density lipoprotein; LDL: low-density lipoprotein cholesterol; HGB: hemoglobin; LEP: leptin; ADPN: adiponectin. In the sit-and-reach test, 0 is the feet line, + is beyond the feet line, − is before the feet line.

**Table 2 biology-13-00567-t002:** Questionnaires scores (mean and standard deviation).

	Mean	SD	Range
SF-36			
PF	66.12	±39.85	70–100
RP	77.93	±17.61	25–100
BP	52.38	±13.54	41–90
GH	65.2	±21.19	30–92
VT	82.98	±21.45	15–90
SF	41.51	±32.35	44.44–100.00
RE	67.28	±21.08	33–100
MH	56.01	±30.58	32–96
HT	54.46	±25.51	25–100
IPAQ			
JOB	1057	±1024.55	330–3816
Transport	1083.16	±1029.71	198–3312
House work	139.82	±231.91	60–1080
Recovery	714.17	±1026.03	23–3876
MET	2642.8	±1864.69	442.2–8544.0
Walk	1402.63	±1447.35	82.5–5841.0
Medium intensity	991.96	±760.62	60–2940
High intensity	779.43	±653.57	120–2520
PSS	43.85	±4.24	34–56
SDS	29.46	±6.87	20–47
TFEQR-21			
Uncontrolled eating	18.46	±4.90	11–31
Restrain	15.07	±4.89	8–26
Emotional eating	12.30	±5.88	4–25
FS-14			
Physical fatigue	3.73	±2.06	2–8
Mental fatigue	2.57	±1.83	1–6
Total score	7.30	±3.26	1–12
PSQI	5.5	±2.70	1–13
HAMA			
Mental anxiety	5.84	±4.64	0–16
Body anxiety	4.69	±2.71	0–12
Total score	10.53	±6.15	2–2618

SF-36: The 36-Item Short Form Health Survey questionnaire; PF: physical function; RP: physical role; BP: bodily pain; GH: general health; VT: vitality; SF: social functioning; RE: emotional role; MH: mental health; HT health changes in the past year; PSS: perceived stress scale; SDS: stress depression scale; IPAQ: international physical activity scale; TFEQR-21: Three Factor Eating Questionnaire-R21; FS-14: Fatigue Scale-14; PSQI: Pittsburgh Sleep Quality Index; HAMA: Hamilton Anxiety Scale.

**Table 3 biology-13-00567-t003:** Correlations of LEP and ADPN with biochemical parameters, anthropometry, functional indices, and questionnaires.

		LEP		ADPN	
		r	*p*	r	*p*
Biochemical parameters	Glucose	0.607	0.001	0.626	0.000
	TC	−0.501	0.000	0.812	0.000
	TG	0.377	0.006	0.547	0.003
	HDL	n.s.	n.s.	0.498	0.007
	LDL	0.602	0.000	n.s	n.s.
	HGB	0.709	0.000	0.545	0.003
	HbA1c	0.283	0.042	n.s.	n.s.
	BMI	0.419	0.026	0.438	0.02
	Hip circum	0.487	0.009	0.471	0.11
Anthropometry	Height	0.531	0.004	0.549	0.002
	Muscle mass	n.s.	n.s.	0.388	0.041
	Fat-free mass	n.s.	n.s.	0.39	0.04
	Biceps control	0.414	0.016	0.474	0.011
	Biceps relaxed	0.491	0.008	0.502	0.006
	Calf circum	0.5	0.007	0.504	0.006
	Shoulder crease skin	n.s.	n.s.	0.384	0.044
	Waist:hip	0.585	0.001	0.608	0.001
	Waist circum	n.s.	n.s.	0.52	0.005
	Thigh circum	0.533	0.003	n.s.	n.s.
Functional indices	1 min-squat (reps)	0.375	0.049	0.416	0.028
	Long jump	0.45	0.016	0.49	0.008
	Hand grip	0.347	0.071	0.398	0.036
	1 min sit-ups	0.362	0.058	0.389	0.041
	Systolic pressure	0.581	0.002	0.599	0.001
	Diastolic pressure	0.597	0.001	0.575	0.002
	METS	0.416	0.028	0.428	0.023
Questionnaires	SDS total	0.47	0.012	0.486	0.009
	PSS total	n.s.	n.s.	0.538	0.003
	SF36/bp	0.546	0.003	0.546	0.003
	SF36/pf	0.379	0.047	0.416	0.028
	PSQI	0.288	0.039	0.538	0.003

TC: total cholesterol; TG: triglyceride; HDL: high-density lipoprotein; LDL: low-density lipoprotein cholesterol; HGB: hemoglobin; SDS: stress depression scale; PSS: perceived stress scale; SF36/bp: bodily pain; SF36/pf: physical functioning; PSQI: Pittsburgh Sleep Quality Index. n.s. = nonsignificant.

## Data Availability

Due to privacy reasons, the datasets used and analyzed during the current study are available from the corresponding author on reasonable request.

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
