# Peer review of "Correlation of Adiponectin and Leptin with Anthropometrics and Behavioral and Physical Performance in Overweight and Obese Chinese College Students"

_biology, 2024, doi:10.3390/biology13080567_

Round 1
Reviewer 1 Report
Comments and Suggestions for Authors
The introduction part of the article is written as it should be.
The small number of participants in the article, the absence of a control group (there is no information about this in the method section), gender inequality (3 females, 23 males), and not stating the time of day when the measurements were made (circadian rhythm) are important shortcomings of the article. Despite this, the study is important and will contribute to the field.
Details of the interventions specifically applied to the participants (nutrition program, physical activity program, sports history, etc.) were not given in the study. Adding this information ensures study reproducibility and understanding of intervention effects.
There are many tables and parameters in the findings section, but the discussion section is written very briefly accordingly. The discussion section needs a little more detail.
The limitations of the article should be written before the conclusion.
Reviewer 2 Report
Comments and Suggestions for Authors
Dear Authors,
The subject is interesting in the area of fat pathophysiology.
It would be better if you had more participants (especially females).
Author Response
Thank you very much for your reviews and suggestions.
Yes, our sample in unbalanced toward males. We added this in the limitation sections:
Our sample is also unbalanced in gender distribution. We are aware the number of participants is limited an umbalanced, but the incidence of obesity in China is much lower than in the Western countries and the sample included almost all the obese students of the Campus.
Reviewer 3 Report
Comments and Suggestions for Authors
- two groups are mentioned in the title: overweight and obese. However, the data is later analyzed jointly for both groups. How many participants were overweight and how many obese? Did the authors consider analyzing the two groups separately? Perhaps analyzing the two groups separately would strengthen some correlations.
- the authors analyzed the two genders together, and it is known that some of the parameters analyzed differ by gender. Did such a combined analysis not distort the observed correlations?
- line 11: explanation of abbreviations should appear earlier
- lines 11-13:this should probably be two sentences
- all references within the text should be changed to comply with the journal's guidelines - "In the text reference numbers should be placed in square brackets"
- line 39: what kind of invasive test?
- line 48: which relationship? From the last sentence? All mentioned above?
- line 61: what does "self-concept" mean in this context?
- line 72: what exactly was randomized in this study?
- line 75: " (tjdxsr046)" - is this the research approval number?
- line 85: one bracket is missing
- line 94: Please provide an example of a serious diabetes complication. So less serious diabetes complications weren`t the problem?
- line 94: what other contraindications?
- line 99: "8 circumferences": chest; minimum waist line, maximum waist line, contracted hip, relaxed hip, contracted thigh, contracted calf, contracted biceps, relaxed thigh, relaxed calf, relaxed biceps - it more than 8
- lines: 110-112: all these parameters were measured with ELISA?
- line 124: how and why was it modified?
- line 130: How vital capacity was measured?
- line 135: how many attempts a participant could make?
- line 148: one bracket is missing
- table 1: can the authors explain how the results of the sit and reach test can be equal -2,90cm?
- table 2: please add the explanation for HT
- line 174: CC - please add the explanation
- line 180,183: shouldn't Table 3 be quoted here?
- line 220: observed in which populations?
- line 245: ".5. Conclusions This section is mandatory, with one or two paragraphs to end the main text." - this should be deleted
Author Response
See attache file.

Round 2
Reviewer 1 Report
Comments and Suggestions for Authors
The authors took my evaluations and recommendations into account. They made the necessary corrections.